# A Lightweight Heuristic for Detecting Unfair Tests in Software Engineering Benchmarks

## Abstract

Software engineering benchmarks are useful tools for evaluating the programming abilities of large language models (LLMs). In addition to ranking models against each other, they can help us to situate the current state of the art by leveraging real-world software engineering problems. For some benchmarks, this latter function is compromised by the presence of 'unfair' tests, meaning tests that contain requirements not specified in the corresponding issue descriptions. Unfortunately, the manual identification of unfair tests is an expensive and time-consuming process; this is especially problematic for automated curation pipelines and continuously-updated benchmarks. There are promising LLM-based solutions, but these come with the usual drawbacks: complex scaffolding, prompt sensitivity, lack of reproducibility and environmental cost; in addition, low recall means the majority of unfair tests are unlikely to be identified. As an alternative to both manual and LLM-based approaches, we propose a lightweight, fully-deterministic, heuristic for the detection of unfair tests in software engineering benchmarks. We evaluate our heuristic against the human annotations used to curate SWE-bench Verified and we compare the results to the corresponding evaluations of two LLM-based alternatives (aligning our methods to facilitate a direct comparison). We find that the accuracy of our heuristic exceeds the accuracy of all non-fine-tuned configurations of both alternatives, but does not exceed the accuracy of a fine-tuned configuration. Given the additional effort, complexity and environmental impact associated with fine-tuning, we consider this to be a positive result. We further propose a version of our heuristic that is less precise, but more sensitive, exceeding the recall of both a fine-tuned, and non-fine-tuned, LLM-based alternative.

## 1 Introduction

Software engineering benchmarks are important tools for evaluating the code generation capabilities of large language models (LLMs). Unlike traditional code generation benchmarks such as HumanEval or MBPP (Chen et al., 2021; Austin et al., 2021), which rely on bespoke programming exercises; software engineering benchmarks collect existing issues from real-world software repositories. This means that software engineering benchmarks are more representative of the challenges associated with real-world software projects; however, it also means they are subject to imperfections in the source repositories and associated software development practices. A good example of this is SWE-bench (Jimenez et al., 2024), a popular software engineering benchmark that has become something of a standard in the field. Whilst the data collection and evaluation frameworks used by SWE-bench are sophisticated and form the basis of many subsequent benchmarks; the benchmark itself contains many task instances that are, effectively, unsolvable due to two key weaknesses in the underlying source repositories:

- **Underspecified issue descriptions**: Many issue descriptions are ambiguous or omit important details about the problem, so no LLM can reasonably be expected to generate a solution.

- **Unfair tests**: Many tests include requirements, such as specific error messages or function names, that are missing from the issue description and cannot be inferred. As a result, no LLM can reasonably be expected to generate a solution that passes the test.

In order to address these weaknesses, Chowdhury et al. developed SWE-bench Verified (SWE-V), a manually-curated variant of SWE-bench that excludes all instances containing underspecified issue descriptions or unfair tests ("unsolvable issues").

## 1.1 Why do we need to exclude unsolvable issues?

Software engineering benchmarks have two distinct functions: to compare the relative performance of different models and to situate the current state of the art. If a substantial portion of the benchmark is unsolvable, the latter function is severely compromised. For example, the highest SWE-bench score is 33.83% at the time of writing,[1] but around 68.3% of task instances evaluated by Chowdhury et al. are marked for exclusion due to unfair tests or unclear issue descriptions (or "other major issues" in some cases). Applying this percentage to the full dataset suggests that SWE-bench may be solved, or nearing completion, but we only know that because of the detailed manual assessment. Without such knowledge, we would have to assume that the remaining issues are solvable and SWE-bench is a long way from completion. This is important because completed benchmarks are no longer useful for determining the current state of the art *or* distinguishing between high-performing models. Additionally, if a benchmark is *nearing* completion, then small improvements might be assigned greater significance (e.g., 99.8% to 99.9% could be viewed as an important change). If the upper limit is unknown, then correctly interpreting absolute scores becomes a much greater challenge.

Unsolvable task instances have already been removed from a a number of software engineering benchmarks, underscoring the need for this type of assessment. SWE-Bench PRO (Deng et al., 2025), Multi-SWE-bench (Zan et al., 2025), SWE-bench Multilingual (Yang et al., 2025) and SWE-Sharp-Bench (Mhatre et al., 2025) use a manual curation process similar to SWE-V, with some variations; SWE-bench-Live (Zhang et al., 2025) and SWE-Rebench (Badertdinov et al., 2025) implement LLM-based filters. The relative merits of manual vs LLM-based approaches are discussed in the following section.

## 1.2 Manual curation and LLM-based approaches

The manual identification of unsolvable task instances is a difficult and time consuming task. For example, in order to curate SWE-V, Chowdhury et al. recruited 93 software developers ("annotators") using a substantial testing and onboarding process; each annotator was provided with a detailed rubric and each SWE-bench instance was labelled by three different annotators before a final score was assigned. A similar process was followed by Zan et al., who recruited 68 developers for the curation of Multi-SWE-bench. Clearly, such investments are not feasible for every research group and represent a substantial time sink. Additionally, there is increasing interest in continuously-updated benchmarks and automated curation pipelines, primarily to mitigate LLM contamination and to generate large numbers of task instances for model training (Zhang et al., 2025; Badertdinov et al., 2025; Adamenko et al., 2025; Guo et al., 2025; Vergopoulos et al., 2025). Since any manual assessment would add significant friction to these pipelines, there is a clear need for automated alternatives.

We have identified two LLM-based tools, SPICE (Oliva et al., 2025) and SWE-Rebench (Badertdinov et al., 2025), that can automatically assess issue clarity and test fairness in software engineering benchmarks. Both these tools are quite complex and go beyond simple LLM prompts. SPICE provides two distinct pipelines: the issue clarity pipeline leverages an LLM to generate a bespoke prompt using recurring patterns of rationales provided by the SWE-V annotators (and extends this with instructions to detect whether issue descriptions include solution code, hints or outlines); the test fairness pipeline uses Aider, a terminal-based programming assistant, to provide repository context to the underlying LLM and an auxiliary LLM to parse Aider's output and retrieve the test fairness label. SWE-Rebench provides a complete pipeline for the construction of software engineering benchmarks, the final stage of which generates issue clarity and test fairness labels similar to SPICE and SWE-V. Unlike SPICE, SWE-Rebench incorporates an LLM that is fine-tuned specifically for the assessment of issue clarity and test fairness.

Both SPICE and SWE-Rebench have been evaluated against the human annotations used to curate SWE-V. Their accuracy is reasonable, suggesting that, in situations where a manual assessment is

---

[1]https://www.swebench.com (*Full* SWE-bench leaderboard as of 3rd December 2025)

infeasible, LLM-based solutions may provide a viable alternative; however, based on the detailed metrics provided by Badertdinov et al. for SWE-Rebench, the recall is quite low indicating significant room for improvement (the recall of SPICE is not reported). These solutions also come with implementation challenges: the prompt engineering infrastructure used by SPICE is sophisticated and requires a third-party programming assistant; SWE-Rebench incorporates a fine-tuned model which would, presumably, need re-tuning for programming languages besides Python. LLMs themselves typically require substantial compute infrastructure or third-party APIs, and their probabilistic nature means that multiple attempts may be required to produce a final result (this is the approach taken by Oliva et al.). Finally, it is important to consider the environmental cost and other ethical concerns associated with LLM inference and training.

### 1.3 Proposed heuristic

As an alternative to both manual and LLM-based solutions, we propose a lightweight, fully-deterministic, heuristic for the detection of unfair tests in software engineering benchmarks. Our basic premise is that any novel lexical elements appearing in the original ("gold") solution patch, and required by the corresponding test, but not mentioned in the issue description, will render the instance unsolvable; effectively, the test contains a hidden specification that is not available to the candidate LLMs. In order to detect such instances, our heuristic takes the intersection of the string, number, and Python identifier tokens in the gold and test patches and compares them to the corresponding issue descriptions; if the intersecting tokens are missing from the issue description, that instance is considered unsolvable and flagged for removal. A detailed description of the heuristic mechanism is provided in Algorithm 1 and Section 2.1.

We evaluate our heuristic by comparing the generated labels to the human annotations used to curate SWE-V and, in order to place our results in context, we compare the resulting accuracy to the accuracies of both SPICE and SWE-Rebench. Since SWE-V annotations were also used to evaluate SPICE and SWE-Rebench, we are able to make a direct comparison by using the same sample instances and aligning our evaluation methodology with Oliva et al. (2025) and Badertdinov et al. (2025). In addition to accuracy, we calculate the recall and precision of our heuristic, alongside various other metrics, and compare these with the detailed metrics provided by Badertdinov et al. for SWE-Rebench. Full details of the evaluation methodology are presented in Section 2.

## 2 Methods

### 2.1 Heuristic mechanism

Algorithm 1 provides a pseudo-code description of our heuristic; this is applied to each instance in SWE-bench or similar benchmarks.

An *instance* in SWE-bench comprises: the issue title and body (the "problem statement"), the solution patch from the pull request that originally resolved the issue (the "gold" patch), and a test patch from the same pull request. Our heuristic has two distinct modes of operation: the "Semantic" mode uses the abstract syntax tree to identify *declared* and *used* Python identifiers in the gold and test patches respectively; the "Tokens-Only" mode uses the lexer to extract *all* Python identifiers from both patches with certain specific exceptions. As such, the Semantic mode is more precise, but potentially less sensitive, than the Tokens-Only mode. String and numeric literals are extracted by the lexer in both modes of operation.

To illustrate the main points of our algorithm, we use the following synthetic instance:

```
    def select_method(dat, method):    + def test_ten():
+    if (method == "ten"):             +    inp = [1,2,3]
+       scale_ten(dat)                 +    exp = [10,20,30]
...                                     +    out = select_method(inp, "ten")
+ def scale_ten(dat):                  +    assert (out == exp)
+    return [x*10 for x in dat]        +    assert (scale_ten(inp) == exp)
```

*Example gold solution patch*       *Example test patch*

---

**Algorithm 1** Heuristic algorithm for detecting unfair tests (in Semantic or Tokens-Only mode)

---

1: **for all** instance $i$ in the benchmark **do**
2:     $S_g \leftarrow$ extract string literals in gold patch of $i$              ▷ Process gold patch
3:     $N_g \leftarrow$ extract numeric literals in gold patch of $i$
4:     **if** Semantic mode **then**
5:         $I_g \leftarrow$ static analysis of relevant *declared* identifiers in gold patch of $i$
6:     **else if** Tokens-Only mode **then**
7:         $I_g \leftarrow$ extract all identifier tokens in gold patch of $i$
8:     **end if**
9:     $S_t \leftarrow$ extract string literals in test patch of $i$             ▷ Process test patch
10:    $N_t \leftarrow$ extract numeric literals in test patch of $i$
11:    **if** Semantic mode **then**
12:        $I_t \leftarrow$ static analysis of relevant *used* identifiers in test patch of $i$
13:    **else if** Tokens-Only mode **then**
14:        $I_t \leftarrow$ extract all identifier tokens in test patch of $i$
15:    **end if**
16:    $S \leftarrow S_g \cap S_t; \;\; N \leftarrow N_g \cap N_t; \;\; I \leftarrow I_g \cap I_t$      ▷ Gold and test patch overlap
17:    $T \leftarrow$ lexical tokens in issue description of $i$
18:    $S_u \leftarrow S \setminus T; \;\; N_u \leftarrow N \setminus T; \;\; I_u \leftarrow I \setminus T$     ▷ Issue description diff ('unspecified')
19:    **if** $(S_u \neq \emptyset) \vee (N_u \neq \emptyset) \vee (I_u \neq \emptyset)$ **then**
20:        flag $i$ for removal
21:    **else**
22:        retain $i$
23:    **end if**
24: **end for**

---

The issue description is simply: *Add new functionality to* `select_method` *for scaling by 10.* Note this does not specify the function name `scale_ten`, nor the string `"ten"` used to select the method.

For the synthetic instance $i$, our algorithm works as follows:

- **Lines 2-8:** From the gold patch, extract the set $S_g$ of all string literals, the set $N_g$ of all numeric literals, and the set $I_g$ of relevant *declared* identifiers (or *all* Python identifiers for the tokens-only mode). For the Semantic mode, constructing $I_g$ requires static analysis of the source files in the patch to extract those bound identifiers which are publicly available.
  For the example, this yields $S_g = \{\texttt{"ten"}\}$, $N_g = \{10\}$, $I_g = \{\texttt{scale\_ten}, \texttt{dat}\}$.

- **Lines 9-15:** From the test patch, extract the set $S_t$ of all string literals, the set $N_t$ of all numeric literals, and the set $I_t$ of relevant *used* identifiers (or *all* Python identifiers for the tokens-only mode). For the Semantic mode, constructing $I_t$ requires analysing the binding structure of the code modified by the test patch in order to determine identifiers that are not already bound.
  For the example, $S_t = \{\texttt{"ten"}\}$, $N_t = \{1, 2, 3, 10, 20, 30\}$, $I_t = \{\texttt{select\_method}, \texttt{scale\_ten}\}$.

- **Line 16:** Compute the intersection of the corresponding sets from the gold patch and test patch.
  For the example, $S = \{\texttt{"ten"}\}$, $N = \{10\}$, $I = \{\texttt{scale\_ten}\}$.

- **Lines 17-18:** Remove any tokens from the overlapping sets that appear in the issue description. Any remaining tokens are considered 'unspecified'.
  For the example, $S_u = \{\texttt{"ten"}\}$, $N_u = \emptyset$, $I_u = \{\texttt{scale\_ten}\}$.

- **Lines 19-23:** Flag the instance for removal (positive result) if there are any unspecified tokens, otherwise retain the instance (negative result). For the example, the condition is true due to underspecification of both the string argument and function name.

## 2.2 IMPLEMENTATION DETAILS

For the current implementation of our heuristic, the string and numeric literals, and Python identifiers in the Tokens-Only configuration, are extracted using the Python tokenizer (the lexer); in the Semantic configuration, Python identifiers, declared in the gold patch and used in the test patch, are extracted using a recursive walk of the abstract syntax tree as outputted by the parser. In both cases, the gold and test patches are first applied to the base commit of the relevant repository and

the modified files and line numbers are passed to the lexer and/or parser as required (such that only tokens in the patched parts of the files are considered). For patches that update multiple files, all the modified files are processed and the resulting token-sets are combined. Modified files that are *not* Python source files are skipped. The final output is a single, binary label for each instance, where true means the instance is flagged for removal. Where processing errors occur that prevent an instance from being assessed, the instance is not flagged for removal so the label is false.

For the *declared* identifiers (line 4 of Algorithm 1), 'relevant' means any declarations or assignments that could feasibly be imported and used by tests contained in a separate module. For Python, this includes: non-nested function names and parameter variable names (which get exported in Python), class names, instance methods and attributes, and global (module-level) variables. Since local variables can be renamed without affecting external imports, these are considered irrelevant (e.g., `inp`, `exp`, and `out` in the example instance). For the *used* identifiers in line 7 of Algorithm 1, 'relevant' means any uses of variables outside of local scopes, i.e., variables that could feasibly have been imported from the ground-truth solution. For both the Semantic and Tokens-Only modes, we allow for standard idiomatic Python names, e.g., `self` or `cls`, and magic method names which are standard across Python objects, e.g. `__init__`, as these would not be unusual and should not need specifying in the issue description. For the Tokens-Only mode, we also allow for Python keywords, soft keywords and built-in Python objects.

Whilst the overall approach taken by our heuristic is not specific to any one programming language, the current implementation is specific to Python as the language of SWE-bench tasks. We have begun the process of incorporating additional language support. The values presented here were calculated using Python 3.12.11 on macOS. Recent experiments have shown that the operation and/or evaluation of our heuristic may be sensitive to the execution environment (both Python / library versions and the operating system); as such, other execution environments may yield lower results or generate additional errors. An anonymised version of the current implementation and associated documentation is provided as supplementary materials.

## 2.3 EVALUATION DATASETS

The following datasets have been used to evaluate our heuristic:

- The SWE-bench dataset on Hugging Face (SWE-bench, 2023).
- The SWE-V annotations published as supplementary materials in Chowdhury et al. (2024).
- The SWE-bench instance identifiers used by Oliva et al. and Badertdinov et al. in their respective evaluations (provided via email).

The SWE-bench dataset contains one instance for each GitHub issue (more precisely, each pull request with a linked issue) in the benchmark, where an *instance* includes: an issue description, a "gold" solution patch, a test patch, and a "base commit" representing the state of the repository prior to patching. Various additional items are included that are not relevant to the work presented here. Full details of the SWE-bench dataset structure can be found on Hugging Face (SWE-bench, 2023).

The SWE-V annotation dataset contains both "raw" and "ensembled" annotations. The *raw* annotations include three annotator records for each SWE-bench instance, with each record containing: a unique annotator ID, an SWE-bench instance ID, and integer scores for issue clarity (known as "underspecified") and test fairness (known as "false negative"). Each score has a value between zero and three inclusive with higher scores resulting in instance exclusions (essentially higher scores are 'worse'). An annotator confidence score, presumably reflecting the subjective confidence of each annotator in their assessment, is also provided. The *ensembled* annotations take the highest-severity (highest) scores for each instance; an overall binary label, determining whether each instance should be excluded, is then derived from a combination of the issue clarity and test fairness scores. Our understanding is that instances are excluded when *either* of the issue clarity or test fairness scores has a value of at least two, or if an annotator has flagged the issue for removal using the "other major issues" flag (which is not considered in this analysis).

In order to facilitate a meaningful comparison of results, Oliva et al. and Badertdinov et al. have kindly provided us with the unique identifiers of the SWE-bench instances used in their respective evaluations. To the best of our knowledge, Oliva et al. use a stratified sample of ten instances from

each SWE-bench repository, excepting those repositories with less than ten instances (in which case all instances are used). Badertdinov et al. use a subset of five repositories, including all instances from those repositories that have corresponding SWE-V annotations and an annotator confidence of four or five (the highest confidence values); instances with missing integer scores for issue clarity or test fairness appear to have been excluded.

Our understanding of the respective evaluation methods of Oliva et al. and Badertdinov et al. are derived from a combination of the relevant literature and email correspondence with the authors, for which we are grateful.

## 2.4 EXPERIMENTS

To calculate the accuracy of our heuristic, we applied it to a sample of SWE-bench instances and compared the generated labels to the human (SWE-V) labels for the same instances. To the greatest extent possible, we aligned our input samples and pre-processing methods to Oliva et al. (2025) and Badertdinov et al. (2025) so the calculated accuracies could be compared directly with the reported accuracies of SPICE and SWE-Rebench. As the two evaluation methods are not identical, this resulted in two different experiments, Experiments 1 and 2. In order to compensate for low inter-rater agreement between SWE-V annotators (Oliva et al., 2025), and varying annotator confidence in the ground-truth labels; we carried out a third experiment, Experiment 3, which enforces full inter-rater agreement and high annotator confidence for all benchmark instances.

For Experiment 1, the SWE-V annotations were re-ensembled by taking the majority (not highest-severity) integer score for test fairness, or the median where no majority exists; a binary test fairness label was then derived by applying an exclusion threshold of 2. For Experiment 2, the raw SWE-V annotations were not ensembled: instead, instances with high annotator confidence values (of 4 or 5) were taken directly from the raw dataset and the existing (binary) test fairness labels were used as the final label; instances with missing integer scores for issue clarity or test fairness were excluded. For Experiment 3, we selected high-confidence annotations from the raw SWE-V dataset using the same criteria as Experiment 2 (confidence values of 4 or 5), and selected task instances where the test fairness scores provided by all three annotators were the same. No ensembling, as such, was required as all three annotators agree; we simply took the agreed-upon score for each instance and derived a binary label using a threshold of 2. We only considered instances with exactly three scores.

For all three experiments, we calculated the accuracy, balanced accuracy, recall, precision, F1, specificity and NPV (negative predictive value) of our generated labels using the processed SWE-V annotations as target labels. For each experiment, both the Semantic and Tokens-Only modes of our heuristic were evaluated, resulting in a total of six experimental configurations. For reference accuracies, we took the highest reported accuracies of SPICE and SWE-Rebench, including both the base and fine-tuned configurations of SWE-Rebench; we also extracted the detailed SWE-Rebench metrics provided in Table 4 of Badertdinov et al. (2025). In order to highlight the extent to which our heuristic, as well as SPICE and SWE-Rebench, add value beyond a simple guess; we calculated a set of random baseline statistics and included these for reference. Since our heuristic doesn't involve any training, as such, that might enable it to learn the class ground-truth dataset; we use a uniform random metric with no weighting and no ability to select the majority class.

The results of Experiments 1, 2 and 3 are presented in Tables 1, 2 and 3 respectively. The confusion matrices for all six experimental configurations are provided in Tables 4 to 9.

## 3 RESULTS

The headline result is that our heuristic is slightly more accurate than both SPICE and SWE-Rebench when they don't use a fine-tuned LLM. Our highest calculated accuracy is 80.2% for the Semantic configuration of Experiment 3; however, it must be noted that all accuracy values are influenced by class imbalance in the ground-truth dataset and this is particularly pronounced for Experiment 3 (74.3% to 25.7% in favour of negative labels). Since class imbalance favours negative labels for all three experiments, the correct *retention* of benchmark instances will be driving accuracy more than the correct *exclusion* of benchmark instances. In order to account for this, we have added a balanced accuracy metric (the average of recall and specificity) to all experiments. We have also calculated a balanced accuracy for SWE-Rebench, which shows that compensating for class imbalance does not

Table 1: Experiment 1 results (SPICE comparison)

| | Random | Heuristic | | LLM Alternative |
| --- | --- | --- | --- | --- |
| | | semantic | tokens-only | |
| Accuracy | 50.0% | 72.7% | 70.9% | 68.5% |
| Balanced | 50.0% | 68.9% | 70.5% | - |
| Precision | 40.0% | 73.3% | 62.5% | - |
| Recall | 50.0% | 50.0% | 68.2% | - |
| F1 Score | 44.4% | 59.5% | 65.2% | - |
| Specificity | 50.0% | 87.9% | 72.7% | - |
| NPV | 60.0% | 72.5% | 77.4% | - |

Table 2: Experiment 2 results (SWE-Rebench comparison)

| | Random | Heuristic | | LLM Alternative | |
| --- | --- | --- | --- | --- | --- |
| | | semantic | tokens-only | base model | fine-tuned |
| Accuracy | 50% | 63% | 62% | 60% | 67% |
| Balanced | 50% | 59% | 61% | 55% | 64% |
| Precision | 44% | 64% | 57% | 76% | 69% |
| Recall | 50% | 34% | 56% | 12% | 42% |
| F1 Score | 47% | 44% | 56% | 21% | 52% |
| Specificity | 50% | 85% | 67% | 97% | 85% |
| NPV | 56% | 62% | 66% | 59% | 66% |

Table 3: Experiment 3 results (reduced ground-truth noise)

| | Random | Heuristic | |
| --- | --- | --- | --- |
| | | semantic | tokens-only |
| Accuracy | 50.0% | 80.2% | 73.3% |
| Balanced | 50.0% | 67.3% | 70.9% |
| Precision | 25.7% | 69.6% | 48.4% |
| Recall | 50.0% | 40.7% | 66.1% |
| F1 Score | 33.9% | 51.3% | 55.9% |
| Specificity | 50.0% | 93.9% | 75.7% |
| NPV | 74.3% | 82.1% | 86.6% |

affect our headline result; that is, the balanced accuracy of our heuristic sits between the balanced accuracy of the non-fine-tuned, and fine-tuned, versions of SWE-Rebench.

**Analysis of additional metrics** In Semantic mode, the recall of our heuristic is lower than, or equal to, random for all three experiments; however, in Tokens-Only mode, the recall is fairly strong for Experiments 1 and 3 at 68.2% and 66.1% respectively (18.2 and 16.1 percentage points above random). For Experiment 2, the recall is only 6 percentage points above random, but this exceeds the recall of SWE-Rebench both with and without fine-tuning (+14 percentage points with fine-tuning and +44 percentage points without). The Tokens-Only F1 score exceeds the SWE-Rebench F1 scores both with and without fine-tuning, but the Semantic-mode F1 score is quite low and only exceeds the non-fine-tuned version of SWE-Rebench (and does not exceed random). The precision of SWE-Rebench is quite high and exceeds the precision of our heuristic both with and without fine-tuning (interestingly, the non-fine-tuned variant of SWE-Rebench is most precise at 76%). As expected, there is a trade-off between precision and recall for the two versions of our heuristic, with the precision of the Semantic mode exceeding the precision of the Tokens-Only mode, and the recall of the Tokens-Only mode exceeding the recall of the Semantic mode, for all three experiments. The very high specificity of the Semantic-mode results suggest that the vast majority of 'fair' tests are correctly labelled in this mode (this may be assisted by the relatively low recall). SWE-Rebench also has a very high specificity.

**Representation of source repositories** It is noted that the task instances used for Experiments 2 and 3 are not distributed equally across source repositories; as such, the properties of over-represented repositories could potentially bias the final results. As an opportunity for further work, we suggest breaking down the results of one or more experiments by repository and evaluating any systematic differences. Depending on the results, future experiments could follow the example of Experiment 1, taking a stratified sample by repository; however, some repositories have very few issues in the ground-truth dataset, so this could substantially reduce the overall sample size.

## 3.1 QUALITATIVE ANALYSIS

The following is a concrete example of a *true positive* result: Instance `astropy-14371`[2] was flagged as an unfair test by the SWE-V reviewers because "*in addition to checking if the atol argument has been added to "is_O3", the tests also check if it has been added to the function "is_rotation", which was never specified in the issue description so any reasonable valid solution will fail*". In Semantic mode, our heuristic computed the intersection of the gold patch (declared) and test patch (used) identifiers as: {`is_O3`, `is_rotation`}. Since the issue description contains the token `is_O3` but not `is_rotation`, our heuristic flagged this instance as unsolvable due to the unspecified identifier `is_rotation`. Additional qualitative analysis, with a focus on *false positive* and *false negative* results, is proposed as an opportunity for further work.

There were a small number of "curation errors", meaning errors that prevented the heuristic from completing the assessment of a given instance. The number of curation errors for each experiment can be found in the confusion matrices (Tables 4 to 9). It should be noted that the number of errors shown in Tables 6 and 7 are inflated due to a deliberate mismatch between the number of raw SWE-V annotations and the number of predicted labels in Experiment 2 (there is often more than one annotation per generated label, as discussed in Section 2.3). There were actually 3 curation errors, out of 152 sample instances, for Experiment 2 in both modes of operation. Across all experiments, the small number of remaining errors appear to result from missing source files, possibly due to files being moved or renamed by the gold or test patches. It is noted that a recent Windows run generated additional errors not present when using macOS; this will be investigated further in due course. All curation errors result in a false label and form a subset of the predicted negatives.

Table 4: Confusion matrix for Experiment 1 (Semantic)

|  | Predicted Positives | Predicted Negatives | Total |
|---|---|---|---|
| Actual Positives | 22 | 22 | 44 |
| Actual Negatives | 8 | 58 | 66 |
| Total | 30 | 80 | 110 |
| Curation Errors | 0 | 1 | 1 |

Table 5: Confusion matrix for Experiment 1 (Tokens-Only)

|  | Predicted Positives | Predicted Negatives | Total |
|---|---|---|---|
| Actual Positives | 30 | 14 | 44 |
| Actual Negatives | 18 | 48 | 66 |
| Total | 48 | 62 | 110 |
| Curation Errors | 0 | 1 | 1 |

## 4 DISCUSSION

The purpose of our heuristic is to identify unfair tests which render benchmark tasks unsolvable, and to mitigate the need for onerous manual interventions. For this purpose, recall is arguably the most important metric and the Tokens-Only version of our heuristic may offer an effective solution. Whilst the precision of the Tokens-Only mode appears low, that does not necessarily mean that a

---

[2]https://github.com/astropy/astropy/pull/14371, issue: https://github.com/astropy/astropy/issues/13694

Table 6: Confusion matrix for Experiment 2 (Semantic)

|  | Predicted Positives | Predicted Negatives | Total |
|---|---|---|---|
| Actual Positives | 61 | 119 | 180 |
| Actual Negatives | 35 | 198 | 233 |
| Total | 96 | 317 | 413 |
| Curation Errors | 0 | 6 | 6 |

Table 7: Confusion matrix for Experiment 2 (Tokens-Only)

|  | Predicted Positives | Predicted Negatives | Total |
|---|---|---|---|
| Actual Positives | 100 | 80 | 180 |
| Actual Negatives | 76 | 157 | 233 |
| Total | 176 | 237 | 413 |
| Curation Errors | 0 | 6 | 6 |

Table 8: Confusion matrix for Experiment 3 (Semantic)

|  | Predicted Positives | Predicted Negatives | Total |
|---|---|---|---|
| Actual Positives | 48 | 70 | 118 |
| Actual Negatives | 21 | 321 | 342 |
| Total | 69 | 391 | 460 |
| Curation Errors | 0 | 0 | 0 |

Table 9: Confusion matrix for Experiment 3 (Tokens-Only)

|  | Predicted Positives | Predicted Negatives | Total |
|---|---|---|---|
| Actual Positives | 78 | 40 | 118 |
| Actual Negatives | 83 | 259 | 342 |
| Total | 161 | 299 | 460 |
| Curation Errors | 0 | 0 | 0 |

high number of 'fair' tests will be erroneously excluded. In fact, for the scenarios presented here, the specificity is generally quite high, suggesting that relatively few task instances are incorrectly flagged; this may be assisted by the class imbalance which favours negative ground-truth labels, and it should be noted class imbalance in the full (ensembled) annotation dataset favours *positive* ground-truth labels. Further analysis is warranted here; but regardless, if the unfiltered dataset is large, which could well be the case for automated curation pipelines, and the final benchmark is designed for evaluation rather than training (i.e., it can be fairly small but needs to be high-quality), then a higher-recall, lower-precision approach may be appropriate. In other circumstances, a higher-precision approach may be needed, which could favour the Semantic mode or an LLM-based alternative (particularly a solution that incorporates a fine-tuned LLM).

The heuristic itself has a number of weaknesses that may provide opportunities for further work. First, we do not include any kind of dynamic analysis: strings that are generated at runtime, e.g., using Python f-strings, will not typically be included in the intersection of solution and test patch tokens; similarly, if a numeric literal in the test is generated dynamically by the solution, it is unlikely to be included in the intersection. We also use a fairly narrow definition of 'unfair test'. The rubric provided to SWE-V annotators asks the question: "Are the tests well-scoped such that all reasonable solutions to the issue should pass the tests?" (Chowdhury et al., 2024). Whilst additional instructions highlight the specific problem of function names, variable names or error messages not being mentioned in the issue description; this does not preclude a broader interpretation of the question itself. It is also worth noting that the current implementation of our heuristic is Python-specific, which could limit its overall reach. However, many software engineering benchmarks are

based on Python and the overall approach of the heuristic is language-agnostic. We have begun the process of incorporating additional language support.

Finally, it is possible that our heuristic has more predictive capability than suggested by this assessment. We have observed that around 90% of task instances flagged for exclusion in the (ensembled) SWE-V annotation dataset contain unfair tests; as such, our heuristic may be able to substantially predict the overall exclusions by predicting unfair tests. We also note a substantial overlap between the issue clarity and test fairness labels: around 83% of instances with low issue clarity also contain unfair tests. We hypothesise that this overlap is, at least in part, due to a real relationship between the two types of exclusion. Further work is proposed to determine if this is the case, and to determine whether any of these observations hold true for other benchmark datasets.

## 5 CONCLUSIONS

Ideally, software engineering benchmarks should provide milestones against which the programming ability of state-of-the-art LLMs can be measured. In order to facilitate this, and to mitigate the need for onerous manual interventions, we have developed a lightweight heuristic to detect unfair tests leading to unsolvable task instances. We have shown that our heuristic is more accurate than all the non-fine-tuned configurations of two LLM-based alternatives, and that one mode of operation is more sensitive than a fine-tuned configuration of one LLM-based alternative. Our heuristic is also deterministic, fully reproducible, and not subject to the practical and ethical challenges associated with LLM inference and training (including environmental impact). Going forward, we plan to explore the prediction of unsolvable task instances as a whole, incorporating both issue clarity and test fairness; we would also like to incorporate multi-language support and distribute our heuristic as a package that can be easily integrated into benchmark curation pipelines.

## 6 REPRODUCIBILITY STATEMENT AND LLM USAGE

An anonymised codebase, containing all necessary code and instructions to reproduce our results, is provided as supplementary materials. With the exception of sample ID lists, which are included in the repository itself (with permission from the authors), the necessary input datasets are downloaded directly from their respective sources. Some pre-processed files are included in the repository and these can be reproduced from the input datasets if needed. All relevant code can be executed using Python3 or Bash. Recent experiments have shown that the operation and/or evaluation of our heuristic may be sensitive to the execution environment (both Python / library versions and the operating system). The values presented here were calculated using Python 3.12.11 on macOS, but it should be noted that other execution environments may yield lower results or generate additional errors.

With the exception of literature search, LLMs have not played any substantial role in the writing of this paper. The use of LLMs for general programming assistance is individual to the authors.

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
