# OpenReview forum: "A Lightweight Heuristic for Detecting Unfair Tests in Software Engineering Benchmarks"
_ICLR.cc/2026/Conference — Submitted to ICLR 2026_

### Official Review · Reviewer_CiXK · 2025-10-15

**Soundness:** 2
**Presentation:** 3
**Contribution:** 1
**Rating:** 2
**Confidence:** 4

**Summary:**

This paper proposes a lightweight heuristic for detecting unfair tests in software engineering benchmarks, a subset of underspecified issues where validation tests rely on information absent from the issue description. The heuristic aims for simplicity, determinism, and reproducibility, and is positioned as an alternative to LLM-based curation methods. The heuristic identifies instances as “unfair” when overlapping string, numeric, or identifier tokens between the test and solution patches are missing from the issue description. Through experiments on SWE-bench and SWE-bench Verified, and by comparison with SPICE and SWE-Rebench, the paper shows that the heuristic achieves accuracy slightly above non–fine-tuned LLM models, though below fine-tuned ones.

**Strengths:**

- The method is simple, deterministic, and reproducible. It avoids the complexity and cost of LLM-based alternatives.
- The empirical evaluation compares directly against two existing automated curation pipelines (SPICE and SWE-Rebench).
- Implementation and datasets are documented and reproducible.

**Weaknesses:**

- The contribution is narrow. The paper solves a subproblem (focusing only on unfair tests, a limited subset of underspecification issues) in an already niche topic (software engineering benchmark curation).
- As a result, the practical utility is unclear: the paper does not convincingly show that unfair tests are a major bottleneck in creating benchmarks, compared to the overall effort, nor that removing them improves benchmark usefulness.
- Comparisons are weak: experiments only compare against random baselines and indirect LLM results. As a side note, a random classifier is always expected to yield 50% accuracy.
- Strong assumptions about variable naming and code quality limit (many valid tests may use inconsistent or auto-generated identifiers).
- Due to mild performance, the proposed heuristic risks flagging many benign instances, which raises anew the need for manual inspection (the very problem that the proposed approach aims to avoid).

**Questions:**

I do not have any specific question whose answer could change my opinion.

---

> ### Author Response · Authors · 2025-11-25
>
> Thank you for your comments, these are addressed below.
>
> **The contribution is narrow. The paper solves a subproblem (focusing only on unfair tests, a limited subset of underspecification issues) in an already niche topic (software engineering benchmark curation).**
>
> Software engineering benchmark curation is a very active field at the moment;
> this is partly due to the danger of contamination from ageing benchmarks that
> pre-date the knowledge cut-offs of state-of-the-art LLMs.
>
> We have carried out a quick analysis showing that around 90% of excluded task
> instances in the (ensembled) SWE-V annotation dataset contain unfair tests; as
> such, our heuristic may largely predict overall exclusions by predicting unfair
> tests. Of course this may be dataset-specific, but we believe it is an encouraging development and may form
> the basis of further work. Additionally, we note that around 83% of instances flagged for low issue clarity are also
> flagged due to unfair tests. This may, at least in part, reflect
> a relationship between the two problems. We have updated Section 4 to address these points and
> propose further work.
>
> **As a result, the practical utility is unclear: the paper does not convincingly show that unfair tests are a major bottleneck in creating benchmarks, compared to the overall effort, nor that removing them improves benchmark usefulness.**
>
> We have added an additional section (Section 1.1) clarifying the motivation
> behind our work. We highlight the fact that, in addition to ranking LLMs
> against each other, software engineering benchmarks are used to situate the
> current state of the field as a whole. If the number of solvable task instances is
> unknown, this functionality is compromised because the absolute scores are
> difficult to interpret. In order to back this up, we have added references to a
> number of benchmarks that have already removed unsolvable benchmark tasks
> (including unfair tests). These are referenced in Section 1.1.
>
> In Section 1.2, we have outlined the considerable effort and expense required to
> identify unsolvable task instances manually. We make
> particular reference to SWE-bench Verified and Multi-SWE-bench who recruited,
> and onboarded, 93 & 68 developers respectively for this task.
>
> **Comparisons are weak: experiments only compare against random baselines and indirect LLM results. As a side note, a random classifier is always expected to yield 50% accuracy.**
>
> To ensure a direct comparison with the LLM-based alternatives, we aligned our
> evaluation methods with theirs and used the same SWE-bench sample instances (which were
> kindly provided by the authors via email). We have clarified this point
> in Sections 2.3 and 2.4. In order to strengthen the comparison, we have
> now incorporated additional reference metrics, including recall and precision,
> for Experiment 2 and updated the analysis and discussion accordingly. Finally, we
> have revised text in Section 2.4 that discussed the random classifier.
>
> **Strong assumptions about variable naming and code quality limit (many valid tests may use inconsistent or auto-generated identifiers).**
>
> I'm afraid we did not fully understand this point. We think
> there are two possible interpretations, which we address as follows:
>
> 1. There are situations where the gold patch _computes_ a string or number
> which corresponds to a literal in the test, in which case the instance is
> unlikely to be flagged. This is discussed in Section 4 and it does
> represent a limitation of our heuristic; however, we do not believe it commonly
> applies to Python identifiers (variable names, function names etc.).
>
> 2. We note that variable names themselves are immaterial since we look at the
> overlap between the gold and test patches. If the latter refers to the former,
> then relevant identifiers will match regardless of their chosen name. For issue
> descriptions, precise naming is required because otherwise LLMs would not be
> able to pass any test that requires the names in question.
>
> **Due to mild performance, the proposed heuristic risks flagging many benign instances, which raises anew the need for manual inspection (the very problem that the proposed approach aims to avoid).**
>
> We have added some notes to Sections 3 and 4 on this point.
> Despite some fairly low precision values, the specificity is generally quite
> reasonable so we don't think a large proportion of benign instances will necessarily
> be excluded (this will depend on the properties of each specific dataset). If
> benign instances _are_ excluded, this is not necessarily a problem so long as
> the final benchmark remains a suitable size for its purpose. As such, we don't
> think manual curation will typically be required in this situation.
> It is noted that existing benchmark curation processes already exclude large
> numbers of issues, and benchmarks are sometimes reduced deliberately in order to
> minimise execution time.

---

### Official Review · Reviewer_K3xD · 2025-10-25

**Soundness:** 2
**Presentation:** 1
**Contribution:** 2
**Rating:** 0
**Confidence:** 2

**Summary:**

This paper proposes a lightweight, deterministic heuristic to identify "unfair tests" in SWE-bench, which is presented as a low-cost, reproducible alternative to expensive, non-deterministic LLM-based curation pipelines.

**Strengths:**

The paper demonstrates a valid test case issue in its methodology section.

**Weaknesses:**

- The paper fails to provide a clear and formal method. In its method section, it shows an example instead of a formal methodology. Additionally, the example seems hard to generalize to other problems for software engineering benchmarks.

- While high-quality test cases are needed, the method proposed in this work is abstract, making it difficult to find real-world test cases that meet the filtering requirements (as question requirements, patch fixes, and test case token distributions are often completely different). Crucially, the paper does not sufficiently justify the filtering, as an unfair test case that causes all models to fail would not affect the relative benchmark scores.

- The paper is densely written and very hard to follow.

**Questions:**

See weaknesses.

---

> ### Author Response · Authors · 2025-11-25
>
> Thank you for your comments, these are addressed below.
>
> **The paper fails to provide a clear and formal method. In its method section, it shows an example instead of a formal methodology. Additionally, the example seems hard to generalize to other problems for software engineering benchmarks**
>
> We have now formalised our approach as a separate algorithm ("Algorithm 1"). We have also redrafted Section 2 to make the overall methodology more clear.
>
> **While high-quality test cases are needed, the method proposed in this work is abstract, making it difficult to find real-world test cases that meet the filtering requirements (as question requirements, patch fixes, and test case token distributions are often completely different). Crucially, the paper does not sufficiently justify the filtering, as an unfair test case that causes all models to fail would not affect the relative benchmark scores.**
>
> Real-world issues from SWE-bench were used to evaluate our heuristic. We have
> added references to a number of real-world benchmarks that apply this type of
> filtering (see Section 1.1), and we have clarified the motivation behind our
> work in this same section. It is noted that, in addition to ranking LLMs against one another,
> software engineering benchmarks are used to determine the current state of the
> art (e.g., "the best LLMs can currently solve 33.8\% of SWE-bench"). If the
> number of solvable task instances isn't known, this functionality is compromised
> because it becomes difficult to interpret the absolute scores.
>
> **The paper is densely written and very hard to follow.**
>
> We have redrafted all sections of the paper to make it clearer and easier to
> read.

---

> > ### Comment · Reviewer_K3xD · 2025-11-25
> >
> > I thank the authors for their response and their efforts in revising the paper. The presentation of the methodology is now much clearer.  In light of these improvements, I have raised my score.

---

### Official Review · Reviewer_t32Q · 2025-10-31

**Soundness:** 2
**Presentation:** 3
**Contribution:** 2
**Rating:** 4
**Confidence:** 4

**Summary:**

This paper introduces a lightweight heuristic method for detecting "unfair tests" in SWE-bench–style benchmarks by comparing identifiers, literals, and AST-extracted tokens between issues and patches. The goal is to identify cases where tests reveal patch content that the model should not have access to. The authors compare the heuristic approach to SWE-bench Verified, SPICE, and SWE-Rebench methods, reporting competitive precision while avoiding LLM inference cost.

**Strengths:**

- Practical problem: test fairness in SWE-bench-style datasets
- Reproducible and deterministic method
- Lightweight — no compute burden unlike LLM curation pipelines
- Good comparison against benchmark curation systems

**Weaknesses:**

- Limited novelty (heuristic token matching)
- Low recall limits real-world usefulness
- Over-generalized claims vs existing LLM curation tools
- Relies on noisy SWE-V labels as "ground truth"
- No hybrid or prompting baselines
- No error analysis or qualitative insights
- Python-only implementation limits generalizability

**Questions:**

1. Why no comparison to a simple GPT-4 zero-shot fairness-classification prompt?
2. Can you report confusion matrix and qualitative cases?
3. How sensitive is performance to project domain or code style?
4. Could hybrid approaches (heuristics + small model) improve recall?
5. Are multi-file patches or cross-file identifiers considered?

---

> ### Author Response · Authors · 2025-11-25
>
> Thank you for your comments and questions, these are addressed below.
>
> **Limited novelty (heuristic token matching)**
>
> We have updated the introduction to clarify the motivation behind our work (see
> Section 1.1 particularly). Whilst token matching is a very general class of
> techniques used elsewhere, we are not aware of any work that uses this approach
> to identify unfair tests in software engineering benchmarks. The only solutions
> to this problem that we know of require large language models or manual curation (which is time
> consuming and expensive).
>
> **Over-generalized claims vs existing LLM curation tools**
>
> We have refined our comparisons with LLM-based approaches throughout the paper.
>
> **Relies on noisy SWE-V labels as "ground truth"**
>
> As mentioned above, we have limited Experiment 3 to SWE-V annotations that have
> full inter-rater agreement and high self-reported annotator confidence, thus reducing the
> ground-truth noise. Further details are provided in Section 2.4 of the revised
> submission.
>
> **No hybrid or prompting baselines**
>
> To ensure a direct comparison with the LLM-based alternatives, we aligned our
> evaluation methods and used the same SWE-bench sample instances. We feel that
> adding an additional, less sophisticated, LLM-based solution would not add sufficient value at
> this time. Regarding the potential for hybrid implementations, please see our
> response to Question 4 below.
>
> **No error analysis or qualitative insights**
>
> We have added a short qualitative analysis section, including an example instance
> and a discussion of errors (see Section 3.1). We hope to extend
> this in due course.
>
> **Python-only implementation limits generalizability**
>
> The overall approach of our heuristic is language-agnostic but the static
> analysis requires some specialisation per language. Our intention is to add
> support for multiple languages; we have already developed a prototype Javascript
> implementation, but it has yet to be evaluated against ground-truth annotations.
>
>
> ## Questions:
>
> **1. Why no comparison to a simple GPT-4 zero-shot fairness-classification prompt?**
>
> The existing LLM-based alternatives include specialised prompt engineering
> pipelines and a fine-tuned LLM; we feel that a zero-shot GPT-4 prompt would be
> inferior to these approaches and wouldn't add sufficient value to the analysis
> at this time. To the greatest extent possible, we aligned our evaluation to the
> LLM-based alternatives by using the same benchmark instances and pre-processing
> methods. This is highlighted in Section 2.4, which has been revised for clarity.
>
> **2. Can you report confusion matrix and qualitative cases?**
>
> The confusion matrices for all experiments are presented in Tables 4 to 9. We
> have added a qualitative case study in Section 3.1 and we hope to extend this
> with additional case studies in due course.
>
> **3. How sensitive is performance to project domain or code style?**
>
> This is a good question. We could break down the results on a per-repository
> basis and evaluate any differences; we have proposed this as further work in
> Section 3. It is noted that Experiment 1, in line with the applicable reference
> project, mitigates any such differences by using the same number of task
> instances for each repository (excepting repositories with very few instances).
>
> **4. Could hybrid approaches (heuristics + small model) improve recall?**
>
> Potentially yes, but one of the benefits of our heuristic is that it mitigates
> the need for large language models and the associated scaffolding etc.. Based on
> the available metrics, our heuristic exceeds the recall of both a fine-tuned and
> non-fine-tuned LLM, so there is no guarantee that a hybrid approach would result
> in a substantial improvement (unless there is relatively little overlap between the two
> sets of positive predictions).
>
> **5. Are multi-file patches or cross-file identifiers considered?**
>
> Yes all the Python source files updated by a patch are included in the analysis.
> We have clarified this point in Section 2.2.

---

### Author Response · Authors · 2025-11-25

We thank the reviewers for all their comments.

We have made the following substantial updates to our project, all of which
are reflected in the revised submission.

**1. Reduced ground-truth noise:** In order to reduce ground-truth noise, we
have limited Experiment 3 to SWE-V annotations that have full inter-rater
agreement and high self-reported annotator confidence. Details of the updated experiment
are provided in Section 2.4 of the revised submission.

**2. Additional reference metrics:** In order to improve the evaluation of our
heuristic, particularly with respect to recall, we have extracted additional
reference metrics from one of the LLM-based alternatives. The additional metrics
are presented in Table 2; the results and discussion sections (Sections 3 and 4)
have been updated accordingly. We have also added a balanced accuracy metric to
account for class imbalance in the ground-truth annotations (which is
particularly pronounced for Experiment 3). Class imbalance is now discussed in
some detail in Section 3.

**3. Paper updates:** We have redrafted most sections to
accommodate reviewer comments and improve overall readability. Most notably, we
have formalised the heuristic mechanism into a separate algorithm (Algorithm 1) and updated Sections 2.1 and 2.2 accordingly. We have also
added a new subsection (Section 1.1) to clarify the motivation behind our work. This
includes additional references to real-world benchmarks that have excluded
unfair tests using both manual and LLM-based approaches.

**4. Reduced errors:** We have resolved a number of processing errors, resulting in some small changes to the results.
The latest results are presented in Tables 1 to 3 and the confusion matrices are presented in Tables 4 to 9.
The remaining errors are discussed in Section 3.1.

We respond to the specific comments of each reviewer below.

---

### Meta-Review · Area_Chair_vVuv · 2026-01-08

**Summary:**

This submission proposes a lightweight, deterministic heuristic to detect "unfair tests" in software engineering benchmarks, positioning it as a low-cost alternative to LLM-based curation. While reviewers acknowledged the benefits of reproducibility and efficiency, they consistently questioned the technical novelty and the narrow scope of the problem.

The rebuttal addressed presentation flaws and ground-truth noise, enabling Reviewer K3xD to raise their score (from 0 to 2). However, all reviewers still maintained negative scores, as the fundamental concerns regarding the limited contribution and unclear practical utility compared to finetuned models remained unresolved.

Therefore, the AC recommends rejection at this stage.

**Reviewer Concerns:**

Addressed:
1. Method Formalization: The authors formalized the heuristic into Algorithm 1 and redrafted the methodology section.
2. Ground Truth Quality: The authors refined the evaluation to use a high-agreement subset of SWE-bench Verified annotations, addressing concerns about label noise.

Unresolved:
1. Limited Novelty: The core contribution remains a simple token-matching heuristic. This lacks significant novelty compared to existing static analysis/ML approaches.
2. Narrow Utility: Argued that the paper solves a subproblem of a niche topic.

**Reviewer Scores:**

- Reviewer t32Q: Remains 4. The reviewer's fundamental concerns were not addressed.
- Reviewer K3xD: 0 to 2. The reviewer acknowledged that the presentation improved. However, the reviewer's fundamental concerns about difficulties in generalizing to the real world were not addressed.
- Reviewer CiXK: Remains 2. This reviewer explicitly stated, "I do not have any specific question whose answer could change my opinion".

---

### Decision · Program_Chairs · 2026-01-26

Reject